# The Effect of Social Capital at the Community and Individual Levels on Farmers' Participation in the Rural Public Goods Provision

**Furong Chen, Yuyuan Yi and Yifu Zhao \***

Institute of Agricultural Economics and Development, Chinese Academy of Agricultural Sciences, Beijing 100081, China; 82101201253@caas.cn (F.C.); 82101202486@caas.cn (Y.Y.)
\* Correspondence: zhaoyifu@caas.cn; Tel.: +86-10-8210-6165

**Abstract:** This study examines the role of social capital, at both individual and community levels, in promoting farmers' participation in providing rural public goods in China. Based on the survey data of 622 farmers from 82 villages across Hebei, Shaanxi, Jiangsu, and Fujian provinces, we used a generalized hierarchical linear model (GHLM) to empirically estimate the effects of social capital on farmers' participation in rural public goods supply. The findings indicate that: (1) community-level factors account for 42.3% of the variance in farmers' participation behavior. The transparency of the public goods construction fund significantly encourages farmers to participate, while the rural collective economy income and the village's geographical location—the distance to the township government—have a significant and negative effect on farmers' participation. (2) On the individual level, social norms, social networks, and social engagement have a positive effect on farmers' participation. The effect of individual social norms is particularly high compared to that of the other two factors. (3) When social capital at the community level is high, the positive effect of individual social networks on farmers' participation is even more significant. Therefore, to encourage farmers to participate in the rural public goods provision, local government should not only pay attention to improving the publicity of public affairs but also cultivate social capital at both the individual and community levels.

**Keywords:** social capital; individual and community; participation; public goods provision; generalized hierarchical linear model (GHLM)

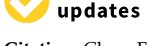



## 1. Introduction

Improving rural public services and infrastructures is essential to elevate rural residents' quality of life and boost rural development. It is also the main pathway to realizing rural revitalization on all fronts and accelerating the construction of agricultural and rural modernization [1,2]. However, the present rural public goods provision system features the government as the main provider leading to many problems in rural public goods provision, such as insufficient supply in quantity, low efficiency, imbalanced supply structure, government financial pressure, poor management and maintenance, etc. [3,4]. Encouraging farmers to participate in the provision is deemed an effective way to ameliorate all these problems—mainly caused by the nearly sole provider system [5]. Farmers' participation in the supply of rural public goods can not only relieve the financial pressure of the government but also solve the problems existing between the supply and demand of rural public goods and offset the "X-efficiency" loss generated by government supply [6,7].

Nevertheless, despite the importance of farmers' active participation in public goods provision, the level of farmers' participation ratio and contribution to public goods provision remains low [5]. According to the survey conducted by Cui (2009) [8], nearly 47.3% of rural residents have not participated in the supply of rural public goods since 2005. When investigating the factors influencing farmers' participation in public goods provision,

researchers have been concerned about the factors at the individual/household and village levels, respectively. From the perspective of individual/household characteristics, the age of the household head, the proportion of non-farm income, being party members or not, being village cadres or not, the level of education, and the perceived living standards are tested as the determinants influencing farmers' participation in the supply of rural public goods [9,10]. From the perspective of the village context, the rural collective economy income, level of urbanization, village topography, and village atmosphere are investigated [11,12].

Besides the individual/household characteristics and village context, social characteristics, social networks, institutional trust, and perceptions can also be potential determinants [13,14]. Due to the characteristic of public goods—non-excludability and non-rivalry—farmers' voluntary participation in public goods provision is subject to the classic problem of free riding, which contributes to collective dilemmas [15,16]. Social capital, conceptualized as features of social life that enable participants to act together more effectively to pursue shared objectives, provides a critical clue to override free riding problems and address collective action dilemmas [17,18]. Generally, social capital can be divided into two levels [19,20]: the individual level, which is formed around individuals and concerned with the ability of individuals to mobilize resources through social networks [21,22], and the community level, which is constructed among residents as a whole within the community [17]. By embedding farmers who pursue maximum individual interests into the social structure through social trust, networks, and norms, individual-level social capital provides an opportunity to overcome collective dilemmas. Community-level social capital constrains individuals' perception of free riding by forming informal regulations, for example, the pressure of moral opinions, through the long-term interactions of residents within the community.

Social capital, therefore, is widely used to analyze individuals' behavior regarding public goods provision. Some studies found that individual trust and mutual help have a positive relationship with their participation in public goods provision [23], and the effect of attitudinal trust is higher than that of behavioral trust in promoting individuals to participate [24]. Some studies found that both individuals' social networks and social trust, in general, are beneficial in promoting farmers' participation in public goods provision, while individuals' special trust and trust in family members, prevent them from participating [6]. Additionally, researchers are also concerned about the effect of community-level social capital on individuals' participation. Unlike the various effects of individual-level social capital on individuals' participation, researchers, in general, reach a consensus that individuals in a community with rich stock of social capital have a stronger willingness to participate in the public goods provision [25]. Community trust [26,27] and community reciprocity [25], particularly, facilitate individuals' participation.

Previous studies have investigated the effect of social capital on individuals' participation in public goods provision and provided precious theoretical references for this paper. However, most previous studies are mainly concerned with the effect at either the individual level or the community level. The few studies that concerned both factors, the individual and the community level, use the traditional liner model, which largely ignores the similarity of individuals within various communities and the differences among communities. A biased estimation result, therefore, could arise [28]. Moreover, studies on the interaction effect between community-level and individual-level social capital on individuals' voluntary participation in public goods provision are limited. Each individual is nested within a particular community, and individuals' decision to participate in rural public goods provision is influenced by both individual features and certain community contexts [12]. The development of individual-level social capital cannot be separated from the community environment, and the enrichment of social capital at the community level may improve individual-level social capital [29].

To promote farmers to participate in public goods provision and bridge the research gap, we, in this study, investigate the effects of two levels of factors, individual and

community, particularly social capital, and their interactions on farmers' participation in public goods provision. Specifically, public goods provision in this paper includes road construction, facilities concerning centralized garbage treatment, domestic sewage treatment, and toilet waste disposal. For that, this paper empirically analyzes the effect with a generalized hierarchical linear model (GHLM) using the survey data of 622 farmers from 82 villages across Hebei, Shaanxi, Jiangsu, and Fujian provinces in China. Our contribution falls into two perspectives. Firstly, we empirically estimated the effect on farmers' participation behavior rather than their willingness or perception, which provides practical references for local government to promote farmers' participation. Secondly, we employed a GHLM model to estimate the effect, which concerns various factors at the community level and contributes to a more precise estimation.

## 2. Theoretical Framework

As an important entry point for overcoming free riding and addressing collective dilemmas, social capital has attracted widespread attention and is regarded as a crucial factor in rural development, especially in rural public goods provision [30,31]. The definition and the measurement of social capital, however, varies across the literature. However, a consensus has been reached that the presence of social capital, including social trust, norms, networks, and engagement, can override free riding incentives and promote cooperation [16,26,32,33]. Farmers' participation in rural public goods provision is a typical collective action. Their decision to participate is not only a result of rational consideration based on individual characteristics but also a result of conformity behavior constrained by social moral opinion pressure within the community [25]. This phenomenon is especially common in rural China, where residents lived in a society based on acquaintance networks and close exchanges. Thus, we explore the influencing mechanism to farmers' participation from individual-level social capital, community-level social capital, and community context. The theoretical influencing path is presented in Figure 1.

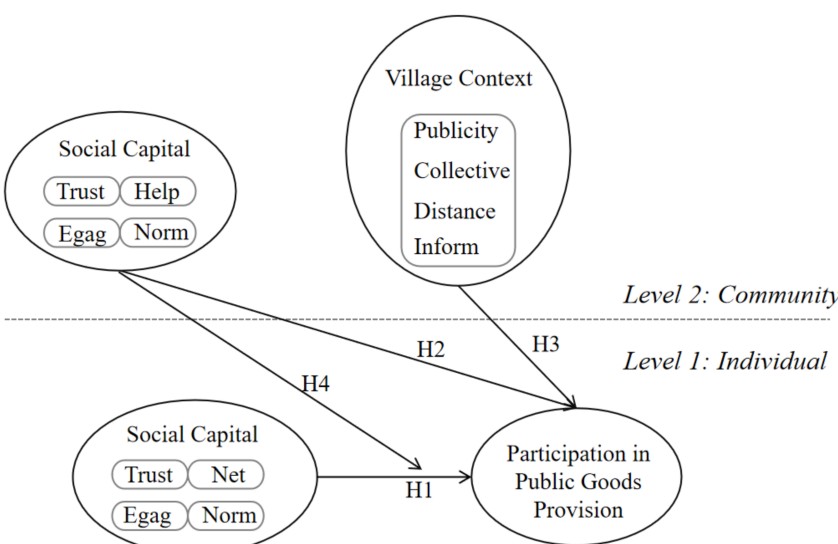

**Figure 1.** Theoretical model.

### 2.1. Individual Social Capital and Farmers' Participation

Taking Putnam's (1993) [26] and Wang's (2021) [34] measurements of individual social capital as references, we measured individual-level social capital using social trust, networks, norms, and engagement. Residents nested in a community have a social network which is formed based on blood, geography, work relationships, social affairs, etc. Individuals make use of the resources carried by other members within the networks and form an information-sharing and distribution mechanism [12]. Effective information exchange

not only improves public engagement's transparency—alleviating the conflict of interests among participants—but also reduces the transaction costs and moral risks of individuals' participation in the public goods provision [35,36].

Individual social trust refers to a particular level of subjective probability with which an agent trusts that another agent would perform a certain action [37]. Individuals with high social trust are optimistic about others' cooperation when promoted to participate, thereby alleviating their concern about others' free riding on public goods provision and contributing to individuals' participation. Meanwhile, individual social trust denotes to what extent individuals are willing to provide credit or rely on the advice of others to take particular actions [38]. Individuals are more prone to participate when they are encouraged by someone trusted.

Individual social norms can be generated through the long-term interaction of all the members in the network of relationships and are regarded as the informal regulations within a certain area of what is obligatory, permissible, or prohibited behavior [39]. Residents living in a community with high social norms are expected to have a high sense of responsibility and honor, which forms a value guide for individuals who are obligated to participate in public goods provision [40].

Besides public goods provision, many community activities also need farmers' participation, such as the election of village cadres, voluntary activities, negotiation, consultation, etc. All of these individual engagements are expected to be associated with farmers' participation in public goods provision. Firstly, individual engagement allows farmers to meet people and widen their social networks, which helps information gathering and exchange. Secondly, individual engagement in community activities allows farmers to express their need for public goods provision and stimulate their initiative and enthusiasm for voluntary participation in collective actions. Thus, we hypothesize the following:

**H1.** *Individual social capital is positively related to farmers' participation.*

**H1a.** *Individual social trust is positively related to farmers' participation.*

**H1b.** *Individual social network is positively related to farmers' participation.*

**H1c.** *Individual social norm is positively related to farmers' participation.*

**H1d.** *Individual social engagement is positively related to farmers' participation.*

### 2.2. Community-Level Social Capital and Farmers' Participation

Social capital at the community level affects farmers' participation in public goods provision as well. According to previous research, social capital at the community level can be measured using social trust, reciprocity, norms, and engagement [41]. Social trust at the community level can effectively reduce the cost of supervision and negotiation in the participation process and increase farmers' propensity to react when village committees call for participation [42,43], thereby promoting farmers to voluntary participation in public goods provision when asked. Social reciprocity at the community level denotes the frequency of mutual help among residents in the community. The more benefits they have received from others in the past, the higher their initial participation inclinations [44]. Unlike individual social norms, community-level social norms affect farmers' participation behavior through external informal incentives and restrictive regulations. Individuals living in a high level of community-level social norms would be more afraid of a reduction in esteem when they act contrary to the norms and more expected to receive rewards spiritually or materially when they obey the rules. The level of community social engagement directly reflects farmers' receptivity to public affairs [45]. High receptivity can encourage farmers' participation in public goods provision. Therefore, we hypothesize the following:

**H2.** *Community-level social capital is positively related to farmers' participation.*

## 2.3. Community Context and Farmers' Participation

Farmers' decisions to participate in public goods provision is not only influenced by farmers' characteristics but also based on certain community contexts [46], such as rural collective economy income, the location of the village, the publicity of public affairs, information accessibility, etc. [12,25]. Rural collective economy income refers to the general income obtained from village collective production and operating activities using collective means of production. A high level of rural collective economy income usually represents the sound development of a village, which means that local governments are more likely to cover all the construction fees for public goods provision and there is less need for farmers to participate. Accordingly, the participation rate would be relatively lower than in a village with a less rural collective economy income. The location of the village may influence farmers' participation as well. Residents in villages near towns have a higher possibility to widen their social network, facilitating information gathering and exchange. Thereby, the greater the distance between the village committee and the town government, the less active farmers' participation might be. The publicity of public affairs assures farmers of the openness and fairness of public funds gathered from individuals, which may improve farmers' confidence in participation and reduce information asymmetry. Information accessibility guarantees farmers awareness of public affairs, which may increase their sense of belonging and ease concerns about public goods provision. Thereby, both publicity and information accessibility may contribute to farmers' voluntary participation. Therefore, we hypothesize the following:

**H3a.** *Rural collective economy income is negatively related to farmers' participation.*

**H3b.** *Distance between the village committee and the town government negatively affects farmers' participation.*

**H3c.** *Information accessibility is positively related to farmers' participation.*

**H3d.** *Publicity is positively related to farmers' participation.*

## 2.4. The Interaction of Community-Level Social Capital and Individual Social Capital to Farmers' Participation

Hypothesis 1 suggests a positive relationship between individual-level social capital and farmers' participation in public goods provision. The relationship is likely moderated by community-level social capital. Community-level social capital has a positive relationship with individual social capital [29]. When a farmer lives in a village with high community-level social capital, they may be actively engaged in collective activities and closely connect with others in the community, which increase their opportunity to develop individual social capital and exchange information with others, and thereby reduce the possibility to free ride on public goods provision. Therefore, we hypothesize the following:

**H4.** *Social capital at the community level strengthens the positive relationship between individual-level social capital and farmers' participation in public goods provision.*

## 3. Materials and Methods

### 3.1. Data Collection and Sampling

Research data in this paper were collected using a questionnaire survey conducted by our research group from June to September 2022 in four provinces of China: Hebei, Shaanxi, Fujian, and Jiangsu. We selected respondents using a stratified random sampling method and conducted the survey via face-to-face interviews. Both individual- and community-level questionnaires were designed and surveyed. The interview object of the individual-level questionnaire was farmers, while the interview object of the community-level questionnaire was village carders. Finally, a total of 650 individual-level and 85 community-level questionnaires were collected. After filtering out the samples with missing key variables, 82 community-level samples and 622 individual-level samples were

valid for our research. This made the ratio of effective samples in this paper 96.47% at the community level and 95.69% at the individual level.

Statistical analysis of the sample is presented in Table 1. Approximately 72.5% of the respondents were male and 27.5% were female. About 66.07% of the respondents were between 35 and 65 years old, which denotes that our samples were mainly in their middle and old age. The education level of respondents was relatively in the middle stage. In total, 38.59% of respondents had finished junior middle school, followed by high or technical school (26.4%) and college and above (22.2%). The average number of family members was 3.54. Our sample households were mainly medium and small; 36.7% were families with 3–4 people and 34.7% with 1–2 persons. Farmers were divided into four groups based on family income: the highest family income group, with a family income in the previous year greater than CNY 100 thousand, accounted for 38.4%, and the group with a family income between CNY 50 and 100 thousand accounted for 32%. As for the participation question, 44.4% of the respondents had participated in the rural public goods provision, meaning that over half of the respondents had not participated.

**Table 1.** Characteristics of sample respondents.

| Variables | Groups | Respondents | Frequency (%) |
|---|---|---|---|
| Gender | Male | 451 | 72.51 |
| | Female | 171 | 27.49 |
| Age | 17–35 | 85 | 13.67 |
| | 35–50 | 187 | 30.06 |
| | 50–65 | 224 | 36.01 |
| | ≥65 | 126 | 20.26 |
| Education | Illiteracy | 11 | 1.77 |
| | Elementary | 69 | 11.09 |
| | Junior high | 240 | 38.59 |
| | High or technical | 164 | 26.37 |
| | College and above | 138 | 22.19 |
| Family size (persons) | 1–2 | 216 | 34.73 |
| | 3–4 | 228 | 36.66 |
| | 5–6 | 151 | 24.28 |
| | ≥7 | 27 | 4.34 |
| Family income (thousand CNY) | ≤10 | 33 | 5.31 |
| | 10–50 | 151 | 24.28 |
| | 50–100 | 199 | 31.99 |
| | ≥100 | 239 | 38.42 |
| Participation | Yes | 276 | 44.37 |
| | No | 346 | 55.63 |

### 3.2. Design of Indicators

Based on the analysis framework and the previous literature, this paper includes determinants from both the individual and community levels.

1.  Dependent variable: Participation (*Part*) is the dependent variable in this study. We measured participation by asking respondents whether they had ever participated in any rural public goods provision by contributing labor, money, or both. If the answer was yes, we assigned participation the value of 1; otherwise, 0.
2.  Individual-level variables: Using Cai's (2015) [6] and He's (2015) [47] research on the factors influencing individuals' participation willingness, we included three types of individual-level factors for this research: individual-level social capital, individual characteristics, and household endowments. Gender, education (*Edu*), and age (*Age*) of individual characteristics and family size (*Fsize*), and family income (*Finc*) of household endowments were selected as control variables. Individual-level social capital (*Isc*) is one of the core independent variables. We measured individual social

capital from four dimensions adopted from previous research [26], individual social trust (*Itru*), networks (*Inet*), norms (*Inorm*), and engagement (*Iegag*). For the individual social trust, we investigated individuals' trust degree to eight objects on a five-point scale (no trust = 1; very trust = 5). For individual social networks, we measured individuals' frequency of communication with six objects with on five-point scale (never = 1; very frequent = 5). The assessment of individual social norms is based on four questions involving moral norms and formal norms. Individual social engagement is measured using four questions concerning vote, negotiation, voluntary affairs, etc. All of these questions are designed to be assessed on a five-point scale. The individual social capital index was measured using the mean value of these four dimensions.

3. Community-level variables: We selected four community-level factors to analyze individuals' participation, namely, rural collective economy income (*Collective*), the location of the village (*Distance*), the publicity of public affairs (*Publicity*), and information accessibility (*Inform*). Social capital at the community level (*Csc*) is another core independent variable in this paper. This variable was measured using four dimensions adopted from previous research [41]: community trust (*Ctru*), reciprocity (*Chelp*), norms (*Cnorm*), and engagement (*Cegag*). Trust at the community level was assessed using farmers' evaluation of their mutual trust. Reciprocity at the community level was assessed via the answer if they agreed that residents in the community help each other a lot. Community norms were measured using their self-evaluation of the obeyance of village regulations and the frequency of getting into trouble with each other. Community engagement was measured via the answer if they agreed that most of the residents in the community participate in collective activity voluntarily. All these questions about community-level social capital were designed as five-point scales (no agree = 1; total agree = 5). We treated the average value of these four dimensions in a specific village as the community-level social capital index. Details of the variables are shown in Table 2.

### 3.3. Method

To examine the proposed model with the nested data, we employed a hierarchical linear model (HLM) to estimate the effect of two levels of factors, particularly social capital, on farmers' participation in public goods provision by HLM 8.1, so as to overcome deficiencies in variance homogeneity and improve the accuracy of model parameter estimation. We assumed that farmers' participation would be explained by both individual- and community-level social capital simultaneously. Since the dependent variable, farmers' participation, is a dummy variable, we evaluated the data using the generalized hierarchical linear model, allowing for the Bernoulli distribution. To improve the interpretability of the intercept and the variance in random intercepts across groups and reduced potential problems associated with multicollinearity, we grand mean centered all the variables except for the categorical variables [48].

#### 3.3.1. Null Model

A null model without predictors is an unconditional mean model which is entered to test the significance of the between-level variance by repeating measurements of dependent variables and dividing the variance into within and between levels [49]. Generally, the statistical significance of the random variance indicates the existence of between-level variance in dependent variables, which means the HLM method is applicable to this study. Additionally, the intraclass correlation coefficient (ICC), which refers to the percentage of variance, can explain how much of the ratio of variance in farmers' participation is at the community level. The equations for this model are as follows.

**Table 2.** The design of indicators and descriptive statistics.

| Variables | Code Value | Mean | SD |
|---|---|---|---|
| | Dependent variable | | |
| Participation (*Part*) | Have you ever participated in public goods provision? Yes = 1, No = 0 | 0.444 | 0.500 |
| | Individual-level variables | | |
| Individual social capital (*Isc*) | The average value of individual social trust, networks, norms, engagement | 4.214 | 0.482 |
| Individual social trust (*Itru*) | The average value of trust score to family members, neighbors, friends, economic elites, social elites, village cadres, outsiders, county government. no trust = 1, less trust = 2, general trust = 3, more trust = 4, very trust = 5 | 4.208 | 0.582 |
| Individual social networks (*Inet*) | The average value of the extent respondents communicate with family members, neighbors, friends, economic elites, social elites, village cadres. never = 1, occasionally = 2, general frequency = 3, more frequent = 4, very frequent = 5 | 3.815 | 0.673 |
| Individual social norms (*Inorm*) | The average value of the answer to the following questions. 1. You are totally familiar with village regulations. 2. You always obey the village regulations. 3. You seldom get into trouble with others. 4. You will always pick up the garbage you notice on the ground. no agree = 1, less agree = 2, general agree = 3, more agree = 4, totally agree = 5 | 4.533 | 0.548 |
| Individual social engagement (*Iegag*) | The average value of the frequency respondents participate in voting for village cadre, negotiating for collective business, others' weddings or funerals, voluntary activities. never = 1, occasionally = 2, general frequency = 3, more frequent = 4, very frequent = 5 | 4.300 | 0.777 |
| Gender | Gender of the respondent: male = 1, female = 0 | 0.725 | 0.447 |
| Age | The actual age of the respondent (years) | 52.529 | 13.838 |
| Education (*edu*) | Education level of the respondent: illiteracy, elementary school, and junior middle school = 0, high school or technical school, college and above = 1 | 0.486 | 0.500 |
| Family size (*Fsize*) | Number of family members (persons) | 3.537 | 1.756 |
| Family income (*Finc*) | The natural log of household income last year | 11.225 | 0.992 |
| | Community-level variables | | |
| Community social capital (*Csc*) | The average value of community reciprocity, engagement, social trust, norms. | 4.330 | 0.320 |
| Community reciprocity (*Chelp*) | Do you agree that people in the community help each other a lot? no agree = 1, less agree = 2, general agree = 3, more agree = 4, totally agree = 5 | 4.477 | 0.304 |
| Community engagement (*Cegag*) | Do you agree that most of the people in the community voluntarily participate in collective activities? no agree = 1, less agree = 2, general agree = 3, more agree = 4, totally agree = 5 | 4.204 | 0.482 |
| Community trust (*Ctru*) | Do you agree that people in the community trust each other? no agree = 1, less agree = 2, general agree = 3, more agree = 4, totally agree = 5 | 4.300 | 0.398 |
| Community norms (*Cnorm*) | The average value of the answer to the following questions. Do you agree that most of the people in the community obey village regulations? Do you agree that people seldom get into trouble with each other in the community? no agree = 1, less agree = 2, general agree = 3, more agree = 4, totally agree = 5 | 4.340 | 0.337 |
| Publicity | To what extent is the fund to construct public goods openness and transparency? Not at all = 1, a little bit = 2, general = 3, almost = 4, totally = 5 | 4.264 | 0.555 |
| Collective economy income (*Collective*) | The collective income of the village last year. ≤50 thousand = 1, 50–200 thousand = 2, 0.2–0.5 million = 3, 0.5–1 million = 4, ≥1 million = 5 | 3.512 | 1.345 |
| Distance | The actual distance between the village committee and county government (kilometer) | 23.091 | 13.317 |
| Information accessibility (*Inform*) | To what extent does the information exchange WeChat group function for communication in the village? Not at all = 1, a little bit = 2, general = 3, almost = 4, totally = 5 | 4.402 | 0.768 |

$$\text{Level 1}: \ \ln(\Pr(Part_{ij} = 1)/1 - \Pr(Part_{ij} = 1)) = \beta_{0j} + r_{ij} \tag{1}$$

$$\text{Level 2}: \ \beta_{0j} = \gamma_{00} + \mu_{0j} \tag{2}$$

where Pr is the probability that the respondent chooses to participate. $Part_{ij}$ represents the participation of the farmer $i$ in community $j$. If they have participated, we assigned participation the value of 1; otherwise, 0. $\beta_{0j}$ is the mean value of participation in the community $j$. $\gamma_{00}$ denotes the intercept of all individuals. $r_{ij}$ is the normally distributed random error term and $\mu_{0j}$ is the residual random effect of community $j$.

### 3.3.2. Random Coefficient Model

According to Bryk and Raudenbush (1992) [50], only if the value of ICC calculated based on the null model is greater than 0.12 is the follow-up of cross-level analysis reasonable. The random coefficient model was used to investigate the effect of individual-level and community-level factors on farmers' participation. Specifically, the equation of the random coefficient model was constructed by entering all the individual- and community-level factors on the basis of the null model (Equations (3) and (4)), which allows the intercept to vary among communities.

$$\text{Level 1}: \ \ln(\Pr(Part_{ij} = 1)/1 - \Pr(Part_{ij} = 1)) = \beta_{0j} + \sum \beta_{ij} x_{ij} + r_{ij} \tag{3}$$

$$\text{Level 2}: \ \beta_{0j} = \gamma_{00} + \sum \gamma_{0q} z_{qj} + \mu_{0j} \tag{4}$$

where $x_{ij}$ is variable at the individual level of individual $i$ in community $j$. $\beta_{ij}$ is the coefficient of dependent variables in level 1. Similarly, $\gamma_{0q}$ and $Z_{qj}$ are the coefficients and variables $q$ at the community level. respectively. The other symbols in Equations (3) and (4) share the same meaning as in Equations (1) and (2).

### 3.3.3. Full Model

To explore the interaction effect of individual-level social capital and community-level social capital on farmers' participation, we construct a full model by adding the community-level factors to the coefficient of individual-level social capital. The model is presented in Equation (5).

$$\text{Level 2}: \ \beta_{pj} = \gamma_{p0} + \sum \gamma_{pq} z_{qj} + \mu_{pj} \tag{5}$$

where $\gamma_{pq}$ is the interaction coefficient between individual social capital and community social capital. $P$ represents the number $p$ individual-level social capital. The model of level 1 in the full model is the same as in the random coefficient model, Equation (3).

## 4. Results and Analysis

### 4.1. Null Model

Before analyzing the effect of individual- and community-level social capital on farmers' participation, a null model is necessary to test the significance of the between-community variance. According to the result estimated by HLM8.1, we found a statistically significant chi-square for farmers' participation ($\chi^2 = 187.89$, $p < 0.001$), indicating a significant between-community variance. To identify the percentage of the variance that resides between clusters, the intraclass correlation coefficient (ICC) was calculated. The result showed that the ICC was 0.423, which suggests that 42.3% of the variance in farmers' participation is at the community level and 57.7% of the variance in farmers' participation is at the individual level. Since ICC > 0.12, the employment of HLM to analyze the effect of individual and community social capital on farmers' participation and the follow-up analysis is reasonable [50].

### 4.2. The Effect of Individual- and Community-Level Factors on Participation

To investigate the effect of both individual and community factors, particularly social capital, on farmers' participation, we use four random coefficient models (Model 1–Model 4)

in this paper. Model 1 includes only general individual-level social capital and individual-level control variables. Model 2 is entered with community-level factors including general community-level social capital based on model 1. To identify the effect of different dimensions of individual-level social capital on farmers' participation, Model 3 and Model 4 replaced general individual-level social capital with four dimensions of individual-level social capital, respectively. Details are presented in Table 3. In general, the between-community variances in all four models were significant at a 1% statistical level, confirming the variance in farmers' participation between community levels.

**Table 3.** Results of random coefficient models.

| Fixed Effect | Model 1 | | Model 2 | | Model 3 | | Model 4 | |
|---|---|---|---|---|---|---|---|---|
| | Coefficient | Stand Error | Coefficient | Stand Error | Coefficient | Stand Error | Coefficient | Stand Error |
| Intercept | −0.285 ** | 0.128 | −0.289 ** | 0.120 | −0.292 ** | 0.129 | −0.294 ** | 0.121 |
| *Isc* | 1.379 *** | 0.223 | 1.355 *** | 0.241 | | | | |
| *Itru* | | | | | −0.081 | 0.203 | −0.119 | 0.204 |
| *Inet* | | | | | 0.433 *** | 0.160 | 0.422 *** | 0.162 |
| *Inorm* | | | | | 0.565 ** | 0.253 | 0.567 ** | 0.256 |
| *Iegag* | | | | | 0.473 *** | 0.168 | 0.513 *** | 0.163 |
| *Edu* | −0.143 | 0.181 | −0.135. | 0.181 | −0.169 | 0.185 | −0.161 | 0.185 |
| *Fpopu* | −0.071 | 0.068 | −0.090 | 0.069 | −0.083 | 0.068 | −0.103 | 0.068 |
| *Gender* | 0.505 ** | 0.248 | 0.508 ** | 0.246 | 0.484 * | 0.250 | 0.479 * | 0.247 |
| *Age* | 0.008 | 0.008 | 0.008 | 0.008 | 0.009 | 0.008 | 0.009 | 0.008 |
| *Finc* | −0.027 | 0.103 | −0.013 | 0.104 | −0.044 | 0.104 | −0.025 | 0.105 |
| *Csc* | | | −0.363 | 0.529 | | | −0.498 | 0.527 |
| *Collective* | | | −0.252 ** | 0.111 | | | −0.285 ** | 0.112 |
| *Distance* | | | −0.018 ** | 0.010 | | | −0.017 * | 0.010 |
| *Inform* | | | 0.132 | 0.119 | | | 0.157 | 0.121 |
| *Publicity* | | | 0.679 ** | 0.260 | | | 0.693 ** | 0.267 |
| Random effect | | | | | | | | |
| Between-Community Variance | 0.730 *** | | 0.578 *** | | 0.768 *** | | 0.600 *** | |
| Within-Community Variance | 0.886 | | 0.894 | | 0.882 | | 0.892 | |

Note: *, **, and *** represent coefficients significant at the 10%, 5%, and 1% statistical levels, respectively.

From the perspective of individual-level factors, individual-level social capital is generally positively associated with farmers' participation in public goods provision ($p < 0.001$). H1 is supported. The odd ratio (Odd ratio = exp (coefficient), which indicates the change in the odd ratio of participation for each unit change in the independent variable) of individual-level social capital in Model 2 (exp (1.379) = 3.876) reveals that the incidence of farmers' participation would increase 3.876 times with one unit of general individual social capital increase, provided other variables do not change. Turning to the different dimensions of individual social capital, we found that all individual social capital has a significant positive relation with farmers' participation, except individual social trust, which has a negative but insignificant effect statistically. H1a is not supported, while H1b–d are supported. According to the results in Model 4, individual social norms promote farmers' participation more than social engagement and social networks. With one unit increase in individual social norms, engagement, and networks, the incidence of farmers' participation would increase 1.763, 1.670, and 1.525 times ((exp (0.567) = 1.763, exp (0.513) = 1.670, exp (0.422) = 1.525), respectively. Concerning individual characteristics, the incidence of males participating in public goods provision is 1.615 times that of females ((exp (0.479) = 1.615).

From the community-level perspective, community-level social capital has a negative but non-statistically significant relation with farmers' participation. H2 is not supported. Among community context factors, rural collective economy income and distance between village committee and county government are significantly negatively associated with farmers' participation in public goods provision. H3a,b are supported. *Inform* shows non-statistically significant and positive effects on farmers' participation. H3c is not supported.

*Publicity*, however, has a significant positive relation with farmers' participation. When publicity increases by one unit, the incident of farmers' participation increases nearly two times accordingly. H3d is supported.

### 4.3. Robustness Test

To test the reliability of HLM results, we conducted the robust test using the logit model based on Fan et al.'s (2022) [51] research. Figure 2 presents the coefficient plot of Model 2 and Model 4 using logit regression. The confidence interval of individual social capital in general, individual social networks, norms, engagement, and publicity at the community level are all above zero, which denotes a significant positive association with farmers' participation. The confidence interval of both rural collective economy income and distance is below zero, indicating a negative effect on farmers' participation. Therefore, the above results are consistent with the HLM results, proving our findings' robustness.

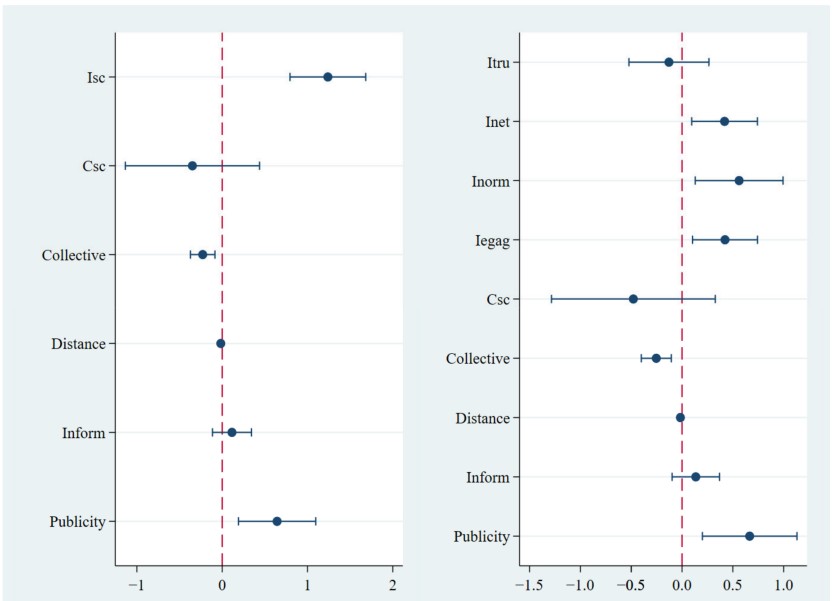

**Figure 2.** Robust test for Models 2 and 4 with the logit model.

### 4.4. The Interaction Effect of Individual- and Community-Level Social Capital on Farmers' Participation

With the aim of investigating whether there is an interaction effect of social capital at the individual and community levels, we constructed five full models by entering cross-level social capital. Both community-level social capital in general (Model 5) and four dimensions (Model 6–9) were investigated and are presented in Table 4. All five models include individual characteristics and community context variables. For simplicity and convenience, only interaction variables concerning social capital are presented.

Hypothesis 4 predicted that social capital at the community level moderates the positive relationship between individual-level social capital and farmers' participation in public goods provision. Table 4 shows that the interaction term of community-level social capital and individual-level network is statistically significant (t = 2.830, $p = 0.005$). H4 is supported. With one unit increase in community-level social capital, the positive effect of individual social networks on farmers' participation would increase 5.4 times, provided no changes occurred in other variables.

Specifically, for the moderate effect of various dimensions of community-level social capital, the interaction term of community-level social reciprocity and individual social networks is positively significant (t = 2.802, $p = 0.005$), indicating that community-level social reciprocity would strengthen the positive effect of individual social network on farmers' participation. However, the development of community-level social reciprocity

hinders the effect of individual social trust and engagement on farmers' participation (t = −2.447, *p* = 0.015; t = −1.937, *p* = 0.053).

**Table 4.** The effect of cross-level social capital on farmers' participation.

| Fixed Effect | | Coefficient | Stand Error | Odd |
|---|---|---|---|---|
| Model 5 | *Csc—Itru* | −0.956 | 0.710 | 0.385 |
| | *Csc—Inet* | 1.689 *** | 0.597 | 5.413 |
| | *Csc—Inorm* | 0.310 | 1.166 | 1.365 |
| | *Csc—Iegag* | −0.697 | 0.686 | 0.498 |
| Model 6 | *Chelp—Itru* | −1.888 ** | 0.772 | 0.151 |
| | *Chelp—Inet* | 1.610 *** | 0.575 | 5.005 |
| | *Chelp—Inorm* | 0.565 | 1.070 | 1.760 |
| | *Chelp—Iegag* | −1.283 ** | 0.662 | 0.277 |
| Model 7 | *Cegag—Itru* | −0.742 | 0.605 | 0.476 |
| | *Cegag—Inet* | 0.981 * | 0.525 | 2.668 |
| | *Cegag—Inorm* | −0.622 | 0.819 | 0.537 |
| | *Cegag—Iegag* | 0.012 | 0.486 | 1.012 |
| Model 8 | *Ctru—Itru* | −0.045 | 0.486 | 0.956 |
| | *Ctru—Inet* | 1.191 ** | 0.501 | 3.290 |
| | *Ctru—Inorm* | 1.176 | 0.754 | 3.243 |
| | *Ctru—Iegag* | −0.914 ** | 0.465 | 0.401 |
| Model 9 | *Cnorm—Itru* | −0.115 | 0.858 | 0.891 |
| | *Cnorm—Inet* | 0.920 * | 0.518 | 2.510 |
| | *Cnorm—Inorm* | −0.394 | 1.045 | 0.674 |
| | *Cnorm—Iegag* | 0.007 | 0.697 | 1.008 |

Note: *, **, and *** represent coefficients significant at the 10%, 5%, and 1% statistical levels, respectively. Odd ratio (odd) = exp (coefficient), which indicates the change in the odd ratio of participation for each unit change in the independent variable.

The interaction term of community-level social engagement and norms with individual social networks are significantly positive (t = 1.867, *p* = 0.062; t = 1.776, *p* = 0.076), indicating that the development of community-level engagement and norms have the potential to strengthen the positive relations between the individual social network and farmers' participation.

Community-level social trust is estimated to have a positive moderate effect on the positive association between farmers' participation and individual social network (t = 2.375, *p* = 0.018). However, social trust at the community level has a negative moderate effect on individual social engagement to promote farmers to participate in public goods provision (t = −1.965, *p* = 0.050).

## 5. Discussion

Social capital at both individual and community levels is expected to have relations with farmers' voluntary participation in public goods provision [2,25,26]. Our research found that individual-level social capital has a significant positive effect on farmers' participation. This result still holds when estimated by the logit model. When individual-level social capital is controlled, the effect of community-level social capital, however, on farmers' participation is non-statistically significant. While social capital at the community level is estimated to have a moderate effect on the positive relationship between individual social capital and farmers' participation, the moderate effect varies from the different dimensions of community-level social capital.

Specifically, except for individual social trust, social networks, norms, and engagement at the individual level are all positively associated with farmers' participation. This is inconsistent with Dai et al.'s (2020) [5] research. Taking the entangled connections with four dimensions of individual-level social capital into consideration [26], the insignificant effect of social trust is reasonable. The development of social trust is based on social networks

or engagement, which therefore can partially explain the information carried by trust. To testify this, we tested the sole effect of individual social trust on farmers' participation, both individual and household characteristics and community context variables controlled, and found that individual social trust has a significant positive effect on farmers' participation, which verified our explanation.

Males are more likely to participate in public goods provision. We thought this might have something related to participation approach. In the questionnaire, we designed a question to gather the information of farmers' participation approaches, and statistical analysis of the participation approaches in different gender groups revealed that the proportion of males choosing to participate by contributing to labor is 13.3% higher than that of females. When both males and females are not willing to contribute money to public goods provision, males are more likely to contribute labor, while females choose not to participate. Another possible reason lies in the fact that females generally have less environmental protection awareness than males [40]. Additionally, our research objectives are mainly focused on environment-related public goods, such as toilets, wastewater, garbage treatment equipment, etc. Males, therefore, have a higher propensity to participate in public goods provision.

As for the community-level factors, the rural collective economy income was found to have a significant negative effect on farmers' participation in public goods provision. With one unit increase in the rural collective economy income, the odd ratio for farmers to participate in public goods provision would reduce by 25%. Statistical analysis on the reason farmers do not participate reveals that 60% of the respondents noted the lack of call for public goods provision in the community. Further analysis found that the rural collective economy income in the group where respondents said there was no call for public goods provision was higher than the group where respondents chose other reasons. Accordingly, we assume that there is less need for a community with a high rural collective economy income to promote farmers' participation in public goods provision, which contributes to the negative effect of the rural collective economy income on farmers' participation. Distance has a negative effect on farmers' participation as well, indicating that farmers living in villages distant from the county government are less likely to participate in the public goods provision. On the one hand, farmers assume that they are strictly supervised and regulated by the county government when they live close to the county government [52], and the procedure of farmers' participation is guaranteed to be fair and transparent, contributing to farmer' voluntary participation. On the other hand, farmers who live in villages with short distances to the county government are more accessible to social network expansion and information exchange by taking advantage of convenient transportation [53], thereby encouraging farmers to participate.

When controlling individual social capital, community-level social capital has a non-statistically significant effect on farmers' participation but moderates the positive effect of individual social capital on farmers' participation. The insignificant effect of community-level social capital on farmers' participation is consistent with Kim's (2018) [54] research, which found that individual-level social networks produce an independent effect on collective actions regardless of community-level social trust. One reasonable explanation is that individual-level social capital can partially represent social capital at the community level. Community-level social capital plays a significant moderate role in the relationship between individual-level social networks and farmers' participation in public goods provision, indicating that the effect of individual social networks on farmers' participation is contingent on community-level social capital.

This study's results clarified that various dimensions of community-level social capital might differ regarding their moderating effect on the relationship between individual social capital and farmers' participation. Community reciprocity is a driver of the positive relations between individual-level networks and farmers' participation, while it is also an inhibitor of the relations between individual-level trust and engagement and farmers' participation. Due to individual social trust's failure to promote farmers to participate,

the moderate effect of community reciprocity on the relationship between individual social trust and farmers' participation cannot be successfully realized. Individual social engagement is expected to increase farmers' propensity to participate by improving their accessibility to broadening social networks and express requirements on public goods construction. However, communities with a high level of reciprocity are prone to form a relatively closed reciprocal network [55], which would prevent farmers from obtaining information and lead to a negative moderate effect.

Engagement and norms at the community level can strengthen the positive association between individual social networks and farmers' participation. Dense individual social networks are able to increase farmers' cost of free riding and promote information exchange, thereby reducing the transaction cost and moral hazard during participation [36]. Farmers in a community with a high level of social engagement find it easy to communicate and exchange information, which is beneficial to widen their individual social networks. The relationship between individual social networks and farmers' participation is, therefore, contingent on community engagement. Meanwhile, high community-level norms are conducive to strengthening the positive relations between individual social networks and farmers' participation. Generally, high community-level social norms denote farmers' stronger sense of self-restraint, responsibility, and collective sense of honor. All of these individual personalities contribute to the effective soft restraint by informal reward and punishment rules developed through dense individual social networks and therefore promote farmers' participation.

The effect of individual social networks and norms can be strengthened by community-level social trust. The existing literature [56] has specified that high community-level trust can effectively reduce the cost of supervision and negotiation during cooperation. Individuals with mutual trust find it easy to establish closed networks. Long-term communication and frequent interaction would not only help strengthen mutual trust but also facilitate the formation of similar behavioral and moral standards among farmers, thus promoting the development of individual social norms. However, the positive relationship between individual engagement and farmers' participation is weakened when community-level social trust is high. One possible reason is that individuals in high community-level trust share information through daily communication, which has the possibility to offset the function of individual engagement for information exchange. Accordingly, community-level trust plays a negative moderate effect on the positive relationship between individual engagement and participation.

## 6. Conclusions and Recommendations

In this study, we explored how to promote farmers to participate in public goods provision from both individual and community level factors, especially social capital, using a generalized hierarchical liner model (GHLM) based on household survey data. Our research found that individual-level social capital, except for individual social trust, has a significant positive effect on farmers' participation, and the effect of social norms is higher than that of social engagement and networks. By adopting the GHLM method, we found that 42.3% of the variance in farmers' participation is at the community level. Specifically, increasing the openness and transparency of the public goods construction fund in a community contributes to farmers' active participation. Communities with a high collective economy income and long distance from the county government have a lower participation rate. When controlling all the factors from the individual and community levels, social capital at the community level has no significant direct influence on farmers' participation but has a positive moderate effect on the relationship between individual-level social capital and farmers' participation in public goods provision.

Based on the above findings, the paper puts forward the following two policy implications on how to motivate farmers to participate in public goods provision. Firstly, local government should focus on the construction of rural culture and develop individual- and community-level social capital. For example, village cadres should formulate village rules

and regulations on the basis of residents' requirements and willingness, so as to improve residents' acceptance and compliance with village rules and regulations, and thereby enhance the constraint effect of social norms on individuals' behavior. To broaden farmers' social network and intensify farmers' connections, efforts can be made by encouraging farmers to participate in various cooperative organizations, which can provide farmers a platform to interact with each other and exchange information. Moreover, facilitating the process of rural digitization and promoting the popularity of rural digital technology in the community could be a potential booster to close farmers' connections. Secondly, improving the openness and transparency of the public goods construction fund in the community benefits farmers' participation as well. To realize that local government should put efforts into establishing a sound democratic consultation and deliberation system, and strictly comply with the regulation of the "4 + 2" system (The "4 + 2" system refers to the democratic policy making process on village affairs under the leadership of village Party organizations. The "4" means four steps: Proposals should be put forward by the Party branch, jointly discussed by the village committee and the Party branch, and deliberated by Party members, and resolutions should be adopted by villagers' representatives; "2" means transparency on two levels—resolutions and implementation results should be made known to the public). Moreover, using modern digital technology tools in the process of rural government is the future trend and would make tremendous progress in enhancing the openness and transparency of public affairs.

**Author Contributions:** Conceptualization, Y.Z.; Data curation, Y.Y. and F.C.; Formal analysis, F.C.; Funding acquisition, Y.Z.; Investigation, Y.Y.; Methodology, F.C. and Y.Y.; Software, F.C.; Supervision, Y.Z. and Y.Y.; Validation, Y.Z.; Writing—original draft, F.C.; Writing—review and editing, F.C., Y.Z. and Y.Y.; All authors have read and agreed to the published version of the manuscript.

**Funding:** This work was supported by The Agricultural Science and Technology Innovation Program of the Chinese Academy of Agricultural Sciences (10-IAED-06-2023) and the Chinese Agriculture Office, Ministry of Agriculture and Rural Affairs soft science project "Study on the rural governance under the background of population outflow" (202216).

**Institutional Review Board Statement:** Not applicable.

**Data Availability Statement:** The datasets used and analyzed during the current study are available from the corresponding author upon reasonable request.

**Conflicts of Interest:** The authors declare no conflict of interest.

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
