# Peer review of "The Effect of Social Capital at the Community and Individual Levels on Farmers’ Participation in the Rural Public Goods Provision"

_agriculture, doi:10.3390/agriculture13061247_

Round 1

Reviewer 1 Report

The proposed text is very interesting and presented in an adequate and fluid manner.

Some parts are perhaps difficult to read (lines 14-24 for example), but in general the exposition is adequate.

 Some clarifications are requested.

The description of what is meant by public goods is absent. A concise review on the subject would be appropriate, but I understand that this is not the focus of the article. However, it would be necessary to point out what the authors are examining.

Another term that should be explained for a non-expert reader is "collective economy". What exactly do the authors mean by this term?

Who exactly was the questionnaire addressed to?  Line 228 mentions rural residents, but then the authors comment by referring to farmers.

Why is the roboustness test only reported for models 2 and 4 in chapter 4.3? Also, the reference to these two models should be stated in the title or in Figure 2 directly.

Chapter 4.4. Why is model 5 applied, if from previous analyses the CSC does not seem statistically relevant, as also reported on line 488?

Chapter 5. The parts between lines 458 and 472 represent information that I cannot find in the previous parts. I may be wrong, but what is the source?

The parts between lines 473-478 seem to repeat considerations already made before.

The recommendation expressed in line 546 does not seem to be addressed in the text above; it should therefore be anticipated in the previous chapters. The same consideration can be made for line 561: the role of social cooperatives does not seem to be explained in the previous text.

In a few other parts, the text is imprecise or unclear.

Chapter 3.2. the first two paragraphs are not numbered, while the third ( comumunity - level variables) is numbered. Also in this paragraph the 4 factors are presented, however “inform” is not then treated (e.g. in the abstract) like the others.

Chapter 4.2 the descriptions of the 4 models is difficult to understand. Table 3 is difficult to read; it should be made clearer, perhaps only by inserting vertical lines.

Chapter 4.4 Table 4 is very difficult to read; it would be advisable at least to introduce lines, separating the models.

The proposed text is very interesting and presented in an adequate and fluid manner. Some parts are perhaps difficult to read, but in general the exposition is adequate.

Reviewer 2 Report

I'd like to see revisions in the regression models and discussion section

The education variable in the regression needs to be recoded., either into binary variables for each level,  or convert into a ratio variable (like illiterate =0, elementary =5, middle school =8, high school =12, college =12)

as of now, college =4x illiteratr, which makes no sense

There is disagreement of using ordinal variables as regressors. I prefer to recode them into one or more binary regressor (e.g., high or low agreement) because they have no unit so marginal effect does not make sense. I encourage this but would not require it

Finally, I'd like to see more detailed description of how closer  farmer connections can be made

There are several grammatical errors and other confusing phrases. I highlighted a few

Reviewer 3 Report

Overall, the manuscript addresses an important and relevant topic, and the findings have implications for policymakers and local governments. The paper is well-organized and interesting. However, there is a need for minor corrections. Below are specific comments and suggestions:

1)Please clarify the “public goods provision” (specify the concrete goods explored in this research) in the Introduction, in brief;

2) Make a detailed presentation of H2 (a, b, c, d) (like H1, H3, according to fig. 1);

3)Clarify the ICC abbreviation (347, 308) correctly;

4) Delete the text repetition (“networks and engagement” - 453).

additional comments:

The methodology of the study is robust and clear. The only thing that could be recommended to improve is the external validity of the results. Given the focus on provinces in China, it is important to acknowledge the limitations in terms of generalizing the findings to other regions or countries. The authors should discuss the external validity of their results and consider the potential applicability and transferability of their findings to different socio-cultural contexts.

The conclusions of the research are consistent with the evidence and arguments presented. The findings outlined in the paper directly contribute to addressing the main question posed. The authors effectively analysed the data and drew meaningful insights pertaining to the impact of social capital on farmers' engagement in rural public goods provision.

The references are appropriate to the research objectives and structure.

It would be better to type an additional line or to place variables’ names not centre-aligned (Table 1). This is also relevant for Table 4. (Model name and Effects) This will help the reader to delineate variables’ groups easily. 

Round 2

Reviewer 2 Report

accept

final editing may be needed